# Single-particle imaging of stress-promoters induction reveals the interplay between MAPK signaling, chromatin and transcription factors

Victoria Wosika [1] & Serge Pelet [1✉]

Precise regulation of gene expression in response to environmental changes is crucial for cell survival, adaptation and proliferation. In eukaryotic cells, extracellular signal integration is often carried out by Mitogen-Activated Protein Kinases (MAPK). Despite a robust MAPK signaling activity, downstream gene expression can display a great variability between single cells. Using a live mRNA reporter, here we monitor the dynamics of transcription in *Saccharomyces cerevisiae* upon hyper-osmotic shock. We find that the transient activity of the MAPK Hog1 opens a temporal window where stress-response genes can be activated. We show that the first minutes of Hog1 activity are essential to control the activation of a promoter. Chromatin repression on a locus slows down this transition and contributes to the variability in gene expression, while binding of transcription factors increases the level of transcription. However, soon after Hog1 activity peaks, negative regulators promote chromatin closure of the locus and transcription progressively stops.

[1] Department of Fundamental Microbiology, University of Lausanne, 1015 Lausanne, Switzerland. ✉email: serge.pelet@unil.ch

A crucial function of all cellular life is the ability to sense its surroundings and adapt to its variations. These changes in the extracellular environment will induce specific cellular responses orchestrated by signal transduction cascades, which receive cues from plasma membrane sensors. This information is turned into a biological response by inducing complex transcriptional programs implicating hundreds of genes[1–3]. Tight regulation of signaling is thus crucial to ensure the correct temporal modulation of gene expression, which can otherwise alter the cell physiology[4–6]. Interestingly, single-cell analyses have revealed that genes regulated by an identical signaling activity can display a high variability in their transcriptional responses[7–10]. This noise in transcriptional output questions how signal transduction can faithfully induce different loci and which molecular mechanisms contribute to the variability in gene expression.

In eukaryotic cells, various environmental stimuli are transduced by the highly conserved Mitogen-Activated Protein Kinases (MAPK) cascades[11,12]. They control a wide range of cellular responses such as cell proliferation, differentiation, or apoptosis. In Saccharomyces cerevisiae, a sudden increase in the osmolarity of the medium is sensed by the High Osmolarity Glycerol (HOG) pathway, which leads to the activation of the MAPK Hog1, a homolog of p38 in mammals[13,14]. Upon hyper-osmotic stress, the kinase activity of Hog1 promotes the adaptation of the cells to their new environment by driving an increase in the internal glycerol concentration, thereby allowing to balance the internal and the external osmotic pressures. In parallel to its cytoplasmic activity, Hog1 also transiently accumulates into the nucleus to induce the expression of hundreds of osmostress-responsive genes (Fig. 1a). The MAPK is recruited to promoter regions by Transcription Factors (TFs) and, in turn, Hog1 recruits chromatin remodeling complexes, the Pre-Initiation Complex, and the RNA Polymerase II (PolII) to trigger gene expression[15,16]. Once cells have adapted, Hog1 is inactivated and exits the cell nucleus, transcription stops, and chromatin is rapidly reassembled at HOG-induced gene loci.

Biochemical analyses of this pathway have identified the key players implicated in gene expression and the central role played by the MAPK in all these steps[15]. In parallel, single-cell measurements have uncovered the large variability present in their expression. In particular, translational reporters and RNA-FISH measurements have identified that slow chromatin remodeling promoted by the MAPK at each individual locus is generating strong intrinsic noise in the activation of many stress-responsive genes[9,17].

In order to get deeper insights into the regulation of osmostress-genes expression kinetics, we aimed at monitoring the dynamics of mRNA production in live single cells. Phage coat protein-based assays, like the MS2 or PP7 systems, have been used to visualize mRNA in live single cells[18–20]. These experiments contributed to revealing the bursty nature of transcription, whereby a set of polymerases simultaneously transcribing a gene generates a burst in mRNA production, which is followed by a pause in transcription[21–23].

In this study, we dissect the kinetics of transcription of osmostress genes. The production of mRNA is monitored using the PP7 phage coat protein assay. This reporter allows us to measure with high temporal resolution and in a fully automated manner, the fluctuations in transcription arising in hundreds of live single cells. This analysis enables to dissect the contribution of various players to the overall transcriptional output. We show that the first few minutes of MAPK activity will determine if a gene is transcribed. We also demonstrate that the chromatin state of a promoter will control the timing of activation and thus the variability in the transcription, while the TF binding will influence the level and duration of the mRNA production.

## Results

**Cellular response to hyper-osmotic stress.** High osmotic pressure is sensed and transduced in the budding yeast Saccharomyces cerevisiae via the HOG signaling cascade, which culminates in the activation of the MAPK Hog1 (Fig. 1a). Upon activation, this key regulator accumulates in the nucleus to trigger gene expression in a stress level-dependent manner (Supplementary Fig. 1a). The activity of the kinase can be monitored by following its own nuclear enrichment[24,25]. In parallel to Hog1, the general stress-response pathway is induced by the hyper-osmotic shock and the transcription factor Msn2 also relocates into the nucleus with dynamics highly similar to the ones observed for Hog1 (Supplementary Fig. 1b, c)[26,27]. Nuclear Hog1 and Msn2 (together with its paralog Msn4) induce osmostress-genes expression, with approximately 250 genes being up-regulated upon osmotic shock[1,28,29]. The activity of the pathway is limited to the cellular adaptation time, which coincides with the nuclear exit of Hog1 and the recovery of the cell size (Supplementary Fig. 1a, d). The fast and transient activity of the osmostress response as well as the homogenous activation of the MAPK within the population[9,25] (Supplementary Fig. 1e), make this signaling pathway an excellent model to understand the induction of eukaryotic stress-responsive genes, which are often accompanied by important chromatin remodeling.

**Monitoring the dynamics of osmostress-genes transcription.** In order to quantify the dynamics of transcription in live single cells, we use the PP7 system to label the production of messenger RNAs (mRNA)[30]. Briefly, constitutively expressed and fluorescently labeled PP7 phage coat proteins strongly associate to a binding partner: an array of twenty-four PP7 mRNA stem-loops (PP7sl). In our settings, this PP7 reporter construct is placed under the control of a promoter of interest and integrated in the genome at the GLT1 locus (Fig. 1b)[30] in a strain bearing a nuclear tag (Hta2-mCherry) and expressing a fluorescently-tagged PP7 protein (PP7ΔFG-GFPenvy[31,32], abbreviated PP7-GFP, Methods). Upon activation of the promoter, local accumulation of newly synthesized transcripts at the Transcription Site (TS) leads to the formation of a bright fluorescent focus due to the enrichment in PP7-GFP fluorescence above the background signal (Fig. 1c and Supplementary Movie 1). The fluorescence intensity at the TS is proportional to the number of mRNA being transcribed and thus to the instantaneous load of RNA polymerases. After termination, single mRNAs are exported out of the nucleus and their fast diffusion in the cytoplasm prevents their detection under the selected illumination conditions.

Typically, time-lapses with fifteen-second intervals for twenty-five minutes with six Z-planes for the PP7-GFP channel on four fields of view were performed. Image segmentation and quantification were performed automatically, allowing to extract up to four hundred single-cell traces for each experiment[33]. The mean intensity of the 20 brightest pixels in the nucleus, from which the median cell fluorescence was subtracted, was used as a measurement of TS intensity and thus as proxy for transcriptional activity (Fig. 1d, Methods).

Figure 1d displays the average TS fluorescence from more than 200 cells bearing the pSTL1-PP7sl reporter following the activation of the HOG pathway by various NaCl concentrations. The HOG-induced STL1 promoter has been extensively studied at the population and single-cell level[9,17,34,35]. As expected, increasing the salt concentration leads to a proportionally increasing transcriptional output from the cell population, whereas no change in TS fluorescence is detected in the control medium.

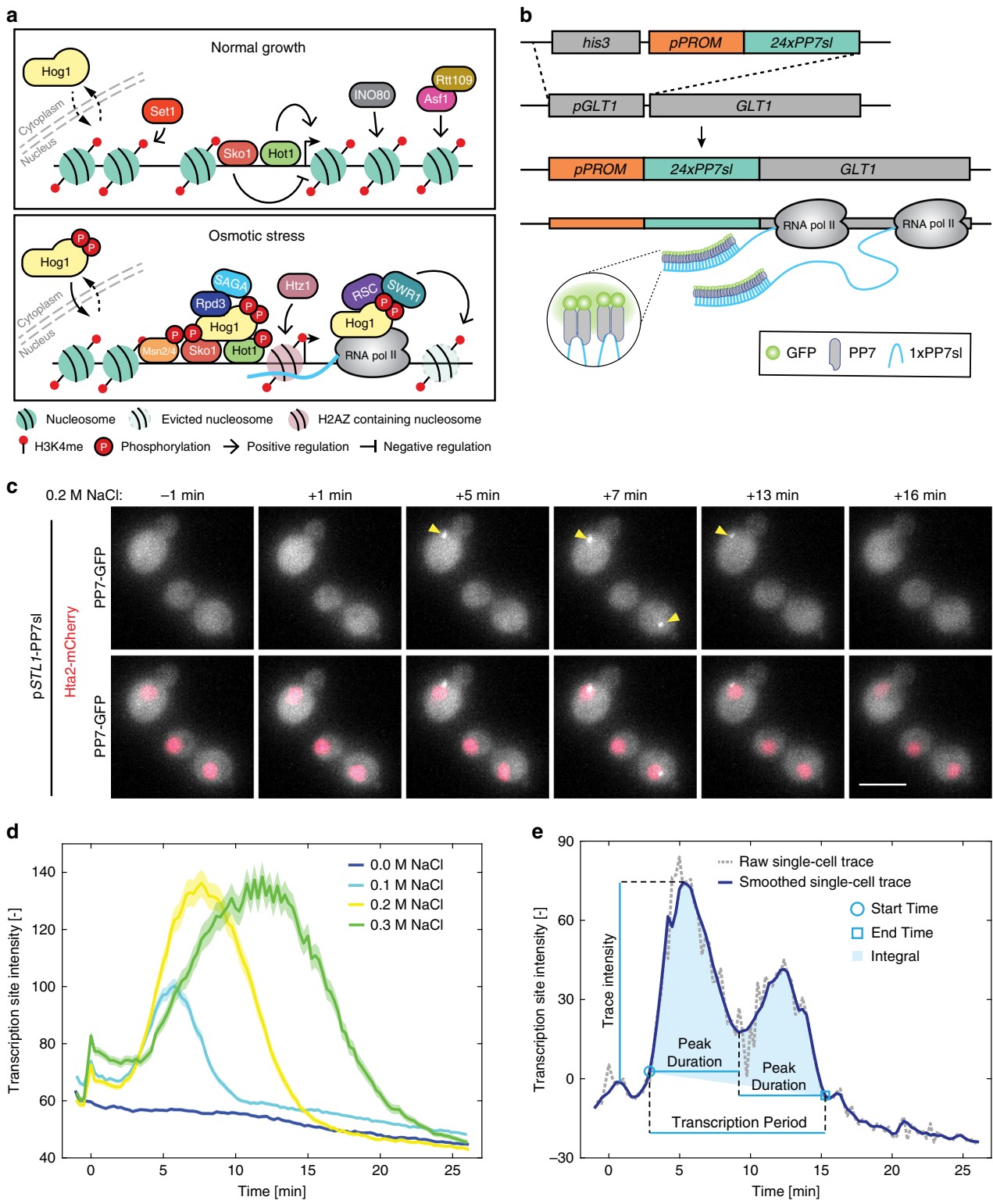

The hundreds of dynamic measurements acquired with the PP7 reporter form a rich dataset where multiple features can be extracted from each single-cell trace (Fig. 1e, Methods). Our automated image segmentation and analysis allow to reliably quantify the appearance (Start Time) and disappearance (End Time) of the TS (Supplementary Fig. 2 and Methods). The maximum intensity of the trace and the integral under the curve provide estimates of the transcriptional output from each promoter (Fig. 1e). In addition, transcriptional bursts can be identified by monitoring strong fluctuations in the TS intensity.

**Validating the live mRNA reporter assay**. The mRNA dynamics measured with the PP7 assay are in close agreement with previously reported data sets[34,36]. Nonetheless, we also verified with a dynamic protein expression reporter that comparable results can be obtained (Supplementary Fig. 3a). The dynamic Protein

**Fig. 1 Monitoring the dynamics of osmostress-genes transcription. a** Schematics of the transcriptional response induced by the MAPK Hog1 upon osmotic stress. Under normal growth conditions, the genomic locus is repressed by histones set in place by the Ino80 complex and Asf1/Rtt109. In addition, H3K4 methylated histones mediated by Set1 contribute to the further repression of the locus (upper panel). When Hog1 is active (lower panel), it accumulates in the nucleus with the transcription factors Msn2/4. Hog1 binds to the transcription factors Hot1 and Sko1, allowing the remodeling of the chromatin by Rpd3 and the SAGA complex. The polymerases can be recruited to the locus and the RSC and SWR complexes evict nucleosomes on the ORF. **b** Construction of the transcriptional reporter. The promoter of interest (pPROM) is cloned in front of 24 stem-loops (24xPP7sl). This construct is transformed in yeast and integrated in the *GLT1* locus 5'UTR, replacing the endogenous promoter. Upon induction of the promoter of interest, the mRNA stem-loops are transcribed and recognized by the fluorescently-tagged PP7 phage coat proteins. **c** Maximum intensity projections of Z-stacks of microscopy images from the p*STL1*-PP7sl reporter system in a 0.2 M NaCl osmotic stress time-lapse experiment. The appearance of bright foci (arrow heads) in the nucleus of the cells denotes the active transcription arising from the promoter. Scale bar represents 5 μm. Representative images from at least three biological replicates. **d** Dynamics of the p*STL1*-PP7sl transcription site intensity (20 brightest pixels in the nucleus minus the median fluorescence of the cell) following hyperosmotic stress. The mean of the population is represented by the solid line. The shaded areas represent the s.e.m. Number of cells for each trace: 0.0 M: 313; 0.1 M: 404; 0.2 M: 229; 0.3 M: 201. **e** Analysis of one representative single-cell trace. The raw trace (gray) is smoothed with a moving average (dark blue) and normalized by subtracting the intensity of the first time point after the stimulus. Multiple quantitative values can be extracted from this trace (see Methods). Source data are provided for **d**.

Synthesis Translocation Reporter (dPSTR) enables to monitor the kinetics of gene expression from a promoter of interest. It bypasses the slow maturation time of Fluorescent Proteins (FP) by monitoring the relocation of the fluorescent signal in the nucleus of the cell[37].

For the PP7 assay, as well as the dPSTR and many other expression reporters, an additional copy of the promoter of interest is inserted in a non-native locus. In order to address if this modified genomic environment alters the dynamics of gene expression, we used CRISPR-Cas9 to integrate the PP7sl downstream of the endogenous *STL1* promoter (Supplementary Fig. 4). Interestingly, we observe only minor differences between the p*STL1* at its endogenous location and at the *GLT1* locus. This observation strongly suggests that the −0.8 kb to TSS of the *STL1* promoter sequence placed at a non-endogenous locus replicates many of the properties of the endogenous promoter.

**Intrinsic noise in osmostress-gene activation**. The microscopy images presented in Fig. 1c illustrate the noise that can be observed in the activation of the p*STL1* promoter upon osmotic stress, which has been previously reported[9,17]. In order to verify that this noise is not due to a lack of activation of the MAPK Hog1 in the non-responding cells, we combined the p*STL1*-PP7sl reporter and the Hog1-mCherry relocation assay in the same strain. As expected, we observe an absence of correlation between the two measurements (Supplementary Fig. 5). Indeed, cells with similar Hog1 relocation behaviors can display highly variable transcriptional outputs.

An additional manner to observe this heterogeneity is to monitor the activation of two *STL1* promoters within the same cell. Using a diploid strain where both *GLT1* loci were modified with either a p*STL1*-24xPP7sl or a p*STL1*-24xMS2sl and expressing PP7-mCherry and MS2-GFP proteins, we observe an uncorrelated activation of both loci within each single cell (Supplementary Fig. 6 and Supplementary Movie 2). This observation confirms the high intrinsic noise generated by the *STL1* promoter upon osmotic stress[9,37]. The highly dynamic measurements provided by the PP7 reporter allows us to decipher some of the parameters that contribute to this large variability.

**High variability in osmostress-genes transcription dynamics**. In addition to p*STL1*, five other stress-responsive promoters often used in the literature to report on the HOG pathway transcriptional activity were selected for this study[34,38]. Each reporter strain differs only by the one thousand base pairs of the promoter present in front of the PP7sl (800 bp for p*STL1*, 660 for p*ALD3*[9,39]); however, each strain displays a different transcriptional response

following a 0.2 M NaCl stimulus (Fig. 2a). Because the level of accumulation of the PP7 signal at the transcription site and the timing of the appearance and disappearance of the TS is different for each tested promoter, it implies that the promoter sequence dictates multiple properties of the transcription dynamics. These dynamic measurements are in general agreement with control experiments performed with the dPSTR assay (Supplementary Fig. 3b) and previously published population-averaged data[34,37]. Importantly, expressing three times more phage coat proteins did not alter substantially the parameters extracted from the PP7 measurements for the two strongest promoters, denoting the absence of titration of PP7-GFP reporter proteins in our experimental settings (Supplementary Fig. 7).

The automated analysis allows to identify the presence or absence of a transcription site in each single cell and thus the fraction of cells that induce the promoter of interest (Fig. 2b). Interestingly, even in absence of stimulus, some promoters display a basal level of transcription. In the p*GRE2*, p*HSP12* and p*GPD1* reporter strains, an active transcription site can be detected in 5–20% of the cells in the few time points before the stimulus (Fig. 2c, Supplementary Movie 3). If the period of observation is extended to a 25-min time lapse without stimulus, this fraction increases twofold to threefold (Supplementary Fig. 8). Upon activation by 0.2 M NaCl, the fraction of responding cells for the three promoters that display basal expression overcomes 85%, while it remains below 65% for the three promoters without basal induction.

**Chromatin state sets the timing of transcription initiation**. A key parameter controlled by the promoter sequence is the timing of induction. In Fig. 2d, the time when cells become transcriptionally active (Start Time) is plotted as a Cumulative Distribution Function (CDF) only for the cells where a TS is detected after the stimulus, thereby excluding basal expressing cells and non-responding cells. Treatment with 0.2 M NaCl results in a sudden activation of transcription (Fig. 2d). This contrasts with non-induced samples, where the CDF of the promoters displaying basal activity rises almost linearly due to stochastic activation during the recording window (Supplementary Fig. 8c).

Upon stress, the promoters displaying basal activity are induced faster than the promoters that are repressed under log-phase growth, with p*GPD1* being activated the fastest (~1 min), while p*ALD3* and p*STL1* require more than 4 min for activation (Fig. 2e). However, there is a great variability in transcription initiation between cells of the same population, since we generally observe 3–4 min delay between the 10th and 90th percentiles, with the exception of p*GPD1* where the induction is more

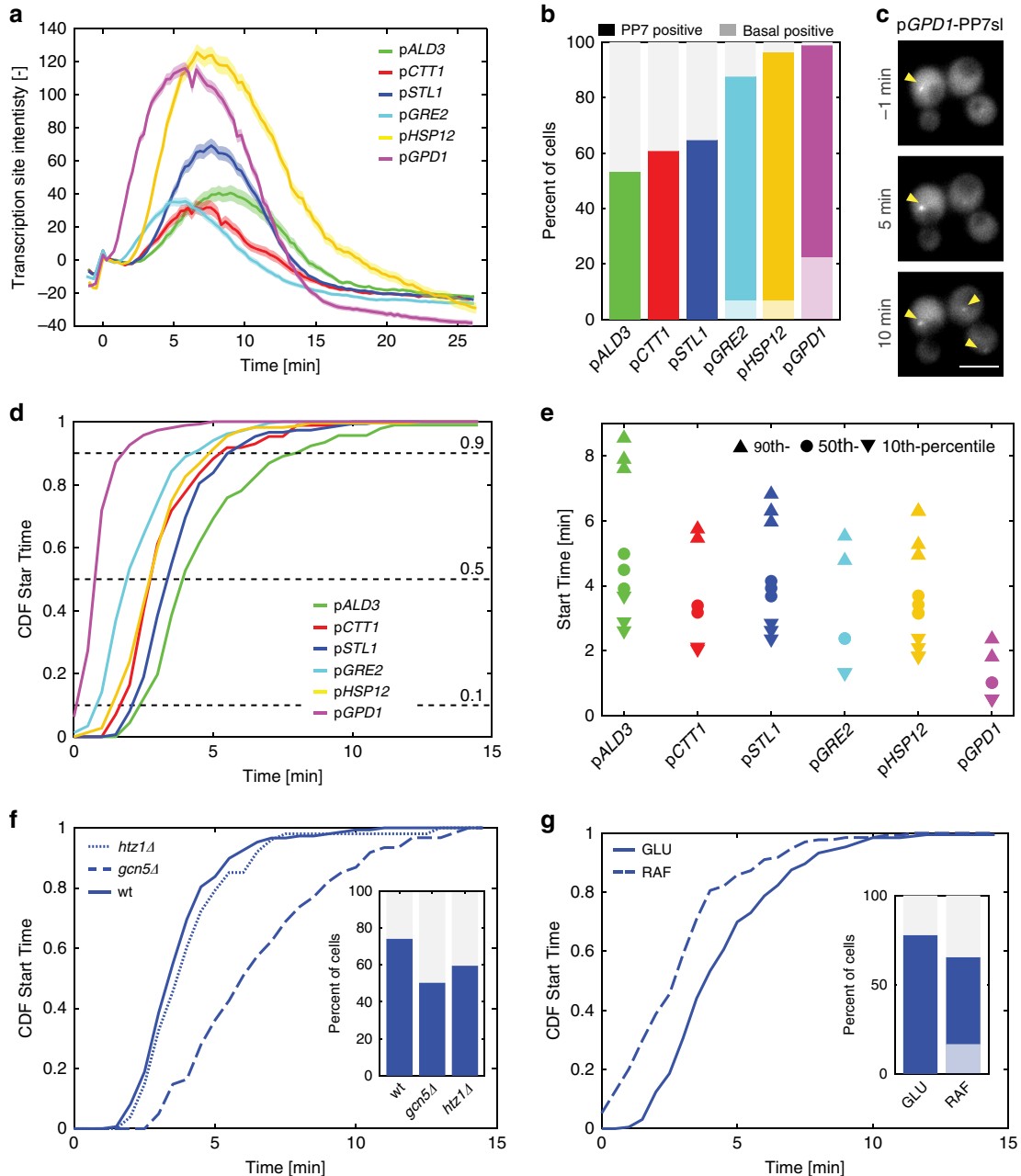

**Fig. 2 Chromatin state dictates the transcription initiation of stress-induced promoters. a** Dynamics of the transcription site intensity from six different promoters following a 0.2 M NaCl stress. The mean of at least 140 cells is represented by the solid line. The shaded areas represent the s.e.m. Number of cells for each trace: pALD3: 171; pCTT1: 140; pSTL1: 229; pGRE2: 289; pHSP12: 243; pGPD1: 335. **b** Percentage of cells where a PP7 TS site was detected. The light shaded area represents the percentage of PP7 positive cells before the stimulus was added (basal transcription). **c** The microscopy thumbnails display cells bearing the pGPD1-PP7sl reporter system, where transcription sites (arrow heads) can be detected before and after the stress of 0.2 M NaCl. Scale bar represents 5 μm. Representative images from at least three biological replicates. **d** Cumulative distribution function (CDF) of the Start Time for each promoter considering only the cells that induce transcription after time zero. **e** 10th, 50th and 90th percentiles of the Start Times shown for the two to three replicates measured for each promoter. **f** Cumulative distribution functions of Start Times for the pSTL1-PP7sl strain in wild type, htz1Δ or gcn5Δ backgrounds. The inset shows the percentage of PP7 positive cells in each background. **g** Cumulative distribution functions of Start Times for the pSTL1-PP7sl strain grown in glucose or raffinose. The inset shows the percentage of PP7 positive cells, the light blue bar the basal positive PP7 cells. Source data are provided for **a**, **f**, and **g**.

uniform and <2 min delay is observed (Fig. 2e). Comparison between individual replicates demonstrates the reliability of our measurement strategy. Interestingly, we observe a positive correlation between faster transcriptional activation from pGPD1, pHSP12, and pGRE2 and the presence of basal expression. These promoters also display the highest numbers of responding cells upon a 0.2 M NaCl shock. These results suggest that basal

expression is associated with a more permissive chromatin state, which enables a faster activation and higher probability of transcription among the cell population.

To test this hypothesis, we disrupted the function of the SAGA chromatin remodeling complex by deleting GCN5[40]. As expected, we observe fewer transcribing cells and a slower induction of the pSTL1 promoter in this background (Fig. 2f). Less remarkably,

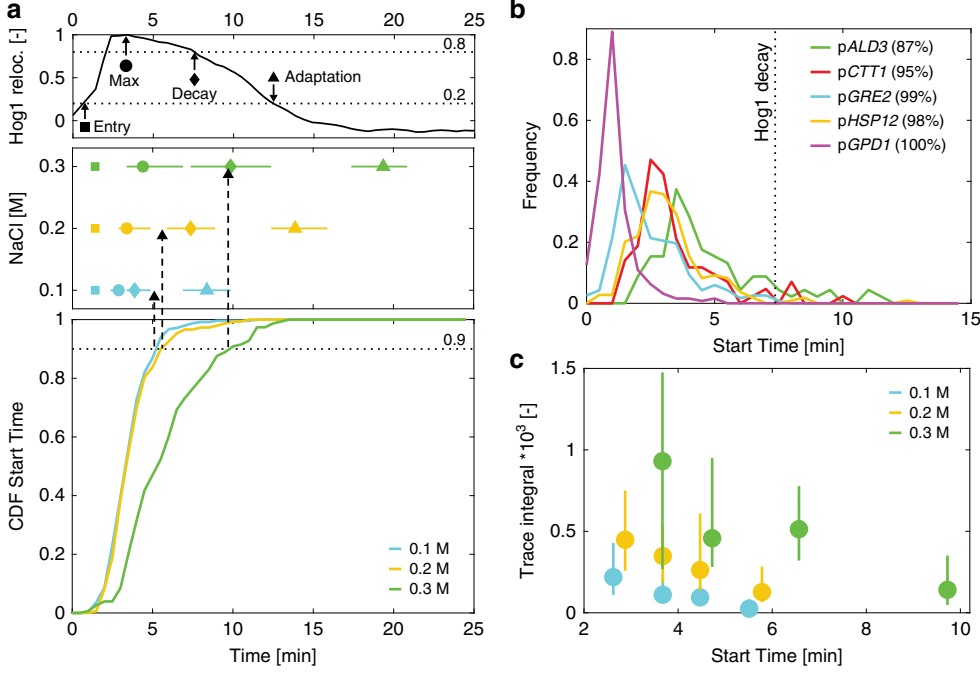

**Fig. 3 Early Hog1 activity dictates promoter activation and output. a** In Hog1 nuclear relocation traces obtained from single cells, the timing of Hog1 nuclear entry (square), maximum enrichment (circle), start of the decay in nuclear enrichment (diamond) and Hog1 adaptation (triangle) can be identified (upper panel). The median (marker) and 25th to 75th percentiles (lines) for these measurements performed on a population of more than 300 cells are plotted for three different osmotic stresses (central panel). The cumulative distribution functions of Start Times for the pSTL1-PP7sl reporter for these same three concentrations are plotted (lower panel). **b** Histograms of Start Times following a 0.2 M stress for the five other promoters tested. The vertical dashed line represents the median decay time of Hog1 measured at 0.2 M. The number in the legend indicates the percentage of cells, which have initiated transcription before the median Hog1 decay time. **c** The population of pSTL1-PP7sl positive cells is split in four quartiles based on their Start Time. The median (circle) and 25th to 75th percentiles (line) of the integral of the PP7 trace is plotted for each quartile of at least 50 cells. Source data are provided for **a**.

abolishing histone H2AZ variants exchange at +1 and −1 nucleosomes by deleting _HTZ1_[41] only results in a reduced percentage of transcribing cells. Conversely, chromatin state at the _STL1_ promoter can be loosened by relieving the glucose repression using raffinose as a C-source[42]. Interestingly, a fraction of the cells grown in these conditions displays basal expression from the pSTL1-PP7sl reporter and the Start Times measured for cells grown in raffinose are accelerated by 1 min compared with the glucose control experiment (Fig. 2g).

The link between the chromatin state under log-phase growth and the ability to induce stress-responsive genes is confirmed by these results. A promoter that is tightly repressed will need more Hog1 activity and thus more time to become transcriptionally active, therefore displaying a lower fraction of responding cells.

**Early Hog1 activity dictates transcriptional competence**. The period of Hog1 activity provides a temporal window, where transcription can potentially be initiated. However, the switch to a transcriptionally active state takes place almost exclusively within the first few minutes after the stimulus. When comparing the characteristic timing of Hog1 nuclear enrichment to the CDF of Start Times for cells bearing the pSTL1 reporter (Fig. 3a and Supplementary Fig. 9a), we observe that 90% of the transcribing cells initiate transcription during the first few minutes of the stress response, while Hog1 nuclear accumulation rises and before it drops below 80% of its maximum (decay time). A similar behavior is observed for all the promoters tested, independently of the presence of basal levels (Fig. 3a, b). For pALD3, which is the slowest promoter tested, 87% of the Start Times are detected before the decay of Hog1 activity (7 min) while the full adaptation time takes 14 min at 0.2 M NaCl.

Interestingly, promoter output also decreases with the time after stimulus. Cells that start transcribing pSTL1 earlier display a larger integral over the PP7 signal and a higher maximum intensity compared with cells that initiate transcription later (Fig. 3c and Supplementary Fig. 9b). A similar behavior is quantified for all tested promoters (Supplementary Fig. 9c, d). These measurements demonstrate that the high Hog1 activity present in the first minutes of the response is key to determine both the transcriptional state and overall output of the promoters.

**TFs control the dynamics and level of mRNA production**. Promoters dictate the timing of transcriptional activation of the ORF and the level at which the mRNA is produced. To extract the transcriptional level of each promoter, we used as a proxy the maximum of the PP7 trace of each single cell where a transcription event could be detected (Fig. 4a). This value represents the maximal loading of polymerases on the locus during the period of transcription. Similar results are obtained when comparing the integral below the PP7 trace, which represents the total transcriptional output from a promoter (Supplementary Fig. 10a). As shown in Fig. 4a, each promoter has an intrinsic capability to induce a given level of transcription, which is independent from the presence of basal transcription or the locus activation time. Indeed, pGRE2 displays the lowest level of induction among all tested promoters, despite the presence of basal transcription and being the second-fastest promoter activated.

As expected, the recruitment of the RNA polymerases is stimulated by the stress; the three promoters with basal activities display a higher transcriptional level upon a 0.2 M NaCl stress than in normal growth conditions (Fig. 4b and Supplementary Fig. 10b). Both the general stress transcription factors Msn2 and

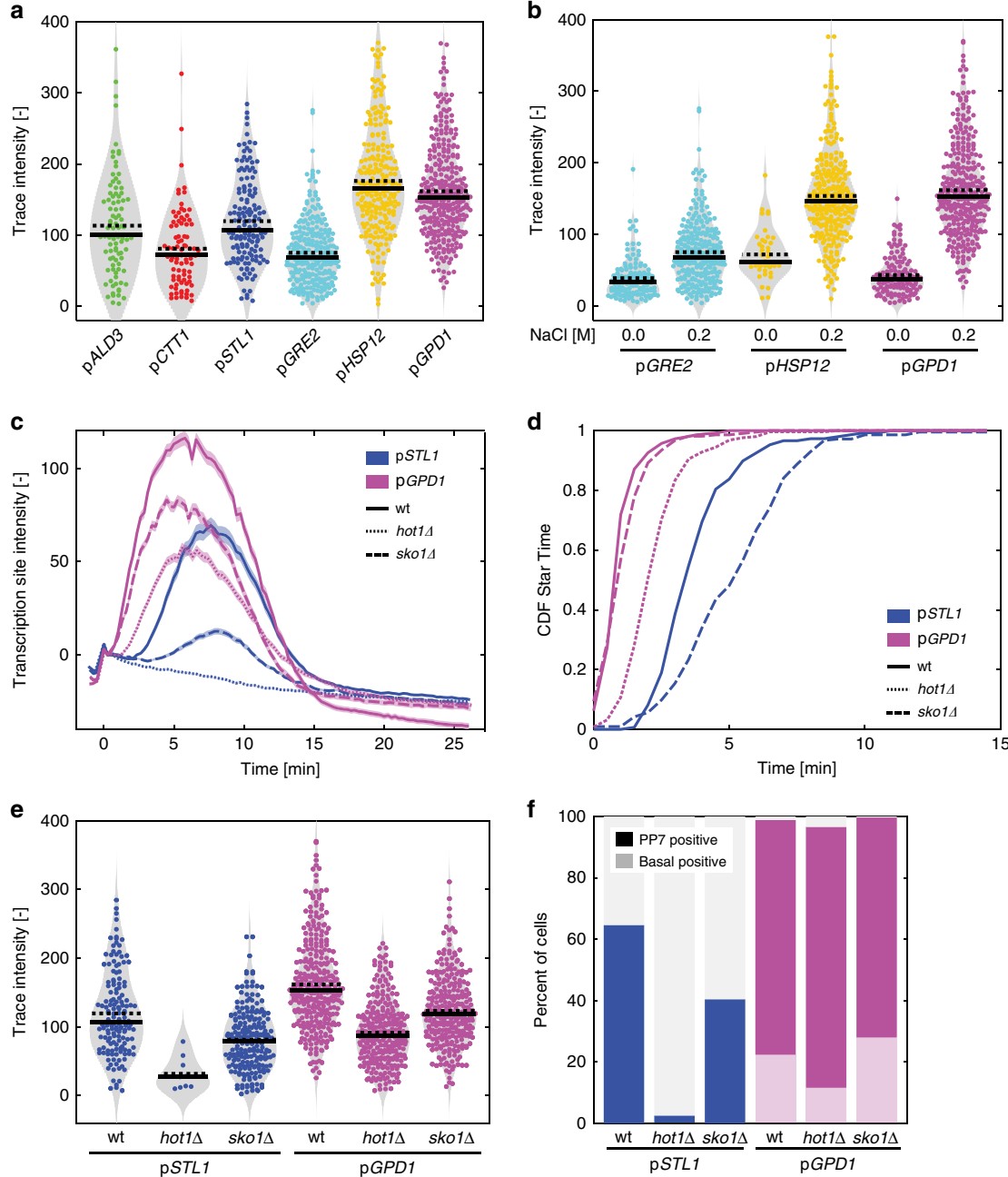

**Fig. 4 Transcription factors control the dynamics and level of mRNA production. a** Violin plots of the trace intensity (maximum of the TS during the transcription period) for the six promoters after stimulation by 0.2 M NaCl. Each dot represents the value calculated from a single cell. The solid line is the median and the dashed line the mean of the population. **b** Comparison between the trace intensity in stimulated (0.2 M NaCl) and unstimulated conditions (0.0 M) for the three promoters displaying basal expression. **c**–**f** Effects of the deletions of the *HOT1* and *SKO1* transcription factor genes on p*STL1* and p*GPD1* dynamics of transcription (**c**, the solid lines represent the mean of the population and the shaded areas represent the s.e.m.), cumulative distribution functions of Start Times (**d**), the trace intensity (**e**) and the percentage of responding cells (**f**) for the p*STL1*-PP7sl and p*GPD1*-PP7sl reporter strains following a 0.2 M NaCl stress for at least 200 cells. For the p*STL1*-PP7sl *hot1Δ* sample, 349 cells were analyzed with only 9 displaying a PP7 TS signal. This low number does not allow to draw a meaningful CDF curve in **d**. Source data are provided for **b** and **c**.

Msn4 and the TFs activated by the MAPK Hog1 (Hot1, Sko1, Smp1) contribute to the transcriptional up-regulation[29,38,43]. Based on studies on synthetic promoters, it has been established that TF binding site number and distance from the Transcription Start Site (TSS) influence the promoter output[44]. Unfortunately, osmostress promoters display a wide diversity in numbers and affinities of TF binding sites and no obvious prediction of the transcriptional activity can be drawn (Supplementary Fig. 11). While multiple Msn2/4 binding sites can be found on the *GPD1*

and *STL1* promoter sequences, their activations are only mildly affected by deletions of these two TFs (Supplementary Fig. 12).

Both *GPD1* and *STL1* are primarily Hog1 targets[28,29,38]. However, their requirements for Hog1 activity is strikingly different. In strains where the MAPK has been anchored to the plasma membrane to limit its nuclear enrichment[45], p*STL1* induction is virtually abolished (only 1.5% transcribing cells), while p*GPD1* activity is barely affected (Supplementary Fig. 13). Similarly, deletion of either TF Sko1 or Hot1 profoundly alter the

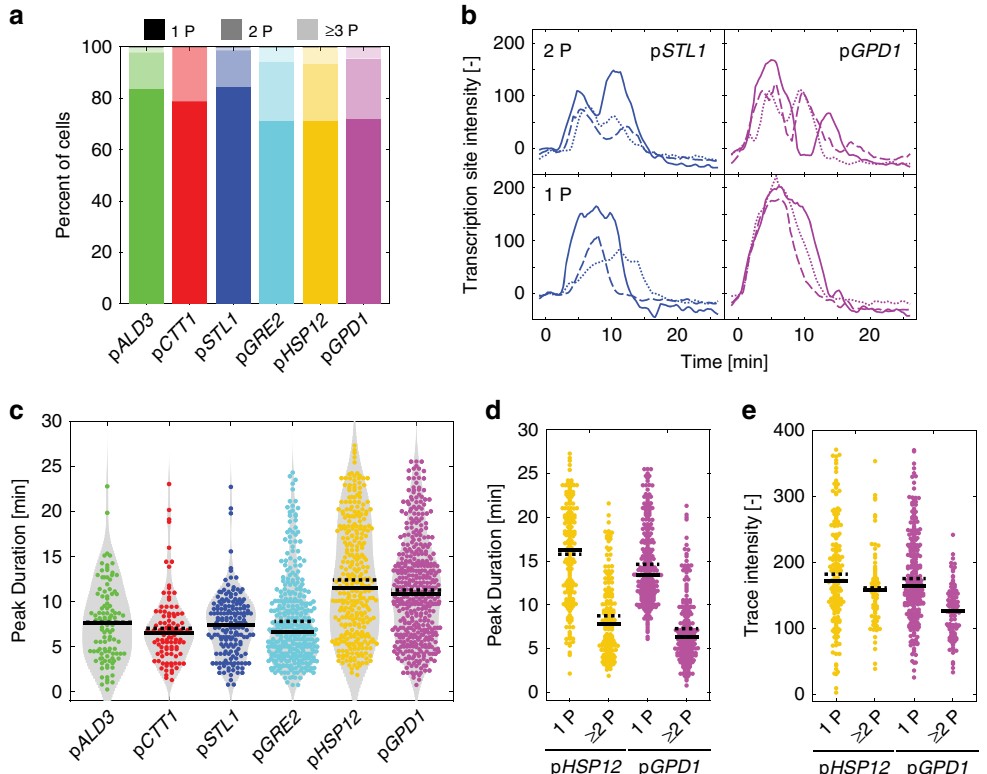

**Fig. 5 Identification of transcriptional bursts in stress-induced transcription. a** Percentage of cells where 1, 2, and 3 or more peaks (1P, 2P, ≥3P, respectively) are identified among the population of responding cells for the different promoters following a 0.2 M NaCl stress. **b** Examples of single-cell traces displaying 1 or 2 peaks (1P, 2P) for the p*STL1*-PP7sl and the p*GPD1*-PP7sl reporter strains. **c** Violin plots representing the Peak Duration. Each dot represents the value calculated for a single peak. The solid line is the median and the dashed line the mean of all the peaks measured. **d–e** The population of cells was split between cells displaying one peak (1P) and two or more peaks (≥2P). The Peak Duration (**d**) and Trace Intensity (**e**) are plotted for the p*HSP12*-PP7sl and p*GPD1*-PP7sl strains. Each dot represents the value calculated for a single peak (**d**) or a single cell (**e**). The solid line is the median and the dashed line the mean of the population.

capacity of p*STL1* to be induced (Fig. 4c–f) while these same deletions have weaker effects on the *GPD1* promoter.

Because the induction of the *STL1* promoter requires an efficient chromatin remodeling, every defect (TF deletion or absence of Hog1 in the nucleus) strongly alters its capability to induce transcription. In comparison, the p*GPD1* is less perturbed by these same defects. These results suggest that TFs act in a cooperative manner on p*STL1*, while they act independently of each other on p*GPD1*.

**Bursts of PolII transcription in osmostress-genes activation.** The PP7 and MS2 systems have allowed to directly visualize transcriptional bursting. In order to identify bursts arising from osmostress promoters, we sought to detect strong fluctuations in each single-cell trace. Fluctuations in TS intensities were filtered to retain only peaks separated by pronounced troughs (Methods). In 20–30% of the traces, two or more peaks are identified (Fig. 5a, b). The total length of the transcript downstream of the promoter is 8 kb (1.5 kb for the stem loops +6.5 kb for *GLT1*). Based on a transcription speed of 20 bp/s[30], the expected lifetime of a transcript at the TS is 6.6 min. This corresponds well to the mean duration observed for the p*ALD3*, p*CTT1*, p*STL1,* and p*GRE2* reporters (Fig. 5c). However, it is unlikely that the strong TS intensities recorded are generated by a single transcript, but rather by a group of RNA PolII that simultaneously transcribe the locus, probably forming convoys of polymerases[46]. Indeed, single mRNA FISH experiments have shown that following a 0.2 M NaCl stress, the endogenous *STL1* locus produces on average 20 mRNAs per cell, with some cells producing up to 100[36].

For p*HSP12* and p*GPD1*, the average peak duration is longer than 11 min (Fig. 5c), suggesting that multiple convoys of polymerases are traveling consecutively through the ORF. Unfortunately, the long half-lives of the transcripts on the locus prevent a separation of individual groups of polymerases. However, when we isolate individual peaks in the single-cell traces, their durations become closer to the expected value of 6.6 min (Fig. 5d). In addition, the output of the transcription estimated by the maximum intensity of the trace, or the integral under the whole curve, is equal or lower for traces with multiple pulses compared with traces where only a single peak is present, indicating a pause in transcription (Fig. 5e and Supplementary Fig. 14). Together these data strengthen the notion that these stress-responsive promoters are highly processive, displaying an elevated rate of transcription once activated. Only brief pauses in the transcription can be observed in a small fraction of the responding cells.

**MAPK activity opens an opportunity window for transcription.** We have shown that transcription initiation is dictated by early Hog1 activity. Next, we want to assess what the determinants of transcription shutoff are and by extension, how the duration of transcriptional activity is controlled. In the HOG pathway, the duration of transcription has been reported to be limited by the cellular adaptation time[34,38]. Therefore, the duration of transcription is shorter after a 0.1 M NaCl stress and longer after a 0.3 M stress, compared with a 0.2 M stress (Fig. 6a). For the *STL1* promoter, the last time point where a PP7 signal is

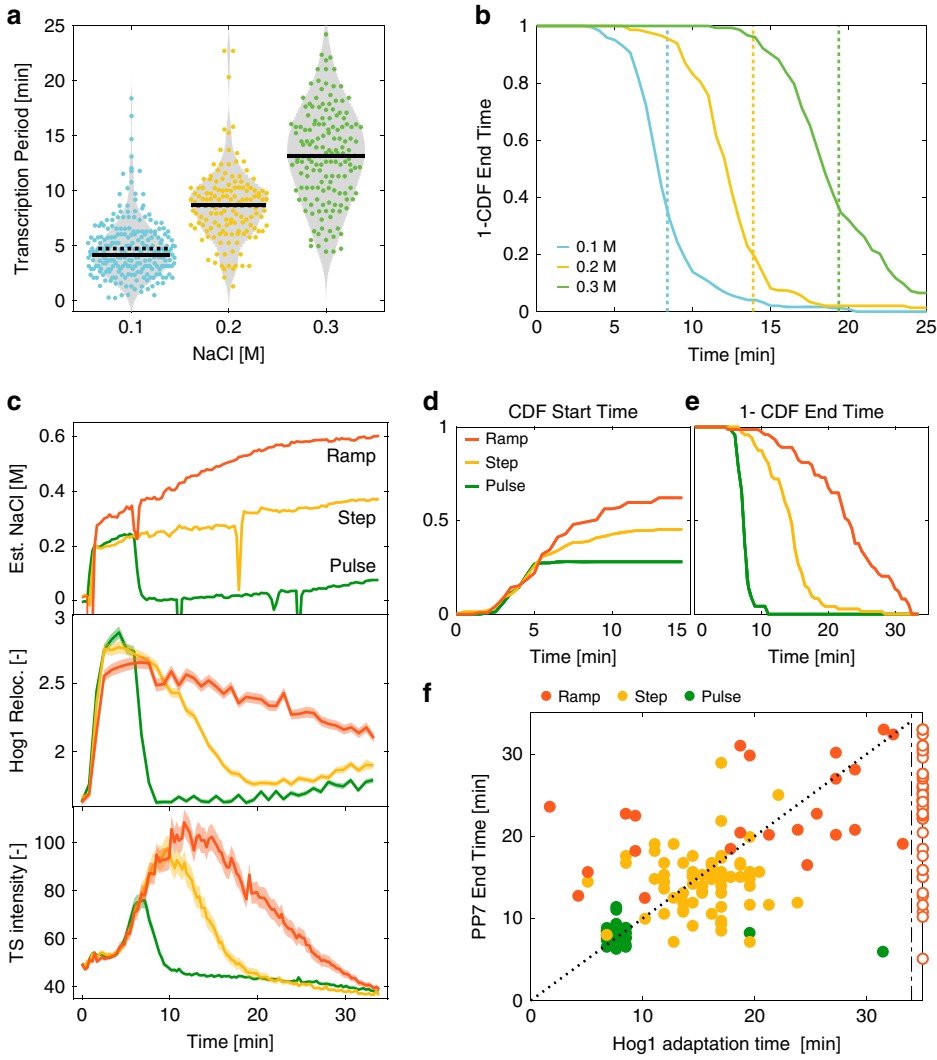

**Fig. 6 Hog1 activity and promoter identity control the shutoff of transcription. a** Violin plots representing the Transcription Period (time difference between End Time and Start Time) measured for the p*STL1*-PP7sl reporter following 0.1, 0.2, and 0.3 M NaCl stresses. Each dot represents the value calculated from a single cell. The solid line is the median and the dashed line the mean of the population. **b** One minus the cumulative distribution function of End Times for the p*STL1*-PP7sl reporter. The vertical dashed lines represent the median adaptation time of Hog1 for the three different stress levels. **c** Dynamics of the estimated NaCl concentrations in the medium for the pulse, step, and ramp experiment protocols (upper panel, Methods). Corresponding Hog1 relocation dynamics (middle panel) and p*STL1*-PP7sl transcription site intensity (lower panel). The mean of at least 180 cells is represented by the solid line. The shaded areas represent the s.e.m. **d** Cumulative distribution function (CDF) of the Start Time for all cells in the pulse, step, and ramp experiments. The CDF at 15 min represents the fraction of responding cells for each condition. **e** One minus the cumulative distribution function of End Times only for the responding cells in the pulse, step, and ramp experiments. **f** Correlation between the Hog1 adaptation time and the PP7 End Time measured in the same cells in the pulse, step, and ramp experiments. The open markers indicate cells where Hog1 has not adapted at the end of the time lapse. Adaptation time is arbitrarily set to 35 min for this sub-population. Source data are provided for **c**.

detected at the TS matches the timing of nuclear exit of the MAPK at all concentrations tested (Fig. 6b).

In order to challenge this link between Hog1 activity and transcriptional arrest, we sought to modulate the MAPK activity pattern by controlling the cellular environment in a dynamic manner. Using a flow channel set-up, we generated a step, a pulse, or a ramp in NaCl concentrations (Fig. 6c, Methods). These experiments were performed in a strain carrying the p*STL1*-PP7sl reporter in conjunction with Hog1-mCherry, allowing to monitor kinase activity and the downstream transcriptional response in the same cell.

The step stimulus at 0.2 M NaCl mimics the experiments performed in wells, where the concentration of the osmolyte is suddenly increased at time zero and remains constant throughout the experiments (Supplementary Movie 4). The mean responses

at the population level (Fig. 6c) confirm this relationship between Hog1 adaptation time and transcription shutoff time. However, at the single-cell level, no direct correlation is observed between these two measurements due to important single-cell variability (Fig. 6f).

In the pulse assay, 7 min after the initial 0.2 M step, the NaCl concentration is set back to 0 M (Supplementary Movie 5). This shortens the MAPK activity period, as Hog1 leaves the nucleus immediately when cells are brought back in normal growth medium. Removing the kinase from the nucleus has a direct impact on the transcriptional process. First, fewer cells become transcriptionally active. Second, the active TS sites disappear within a few minutes after the end of the pulse (Fig. 6d, e). Therefore in this context, we observe a direct correlation between Hog1 activity and transcription, which is in line with the known

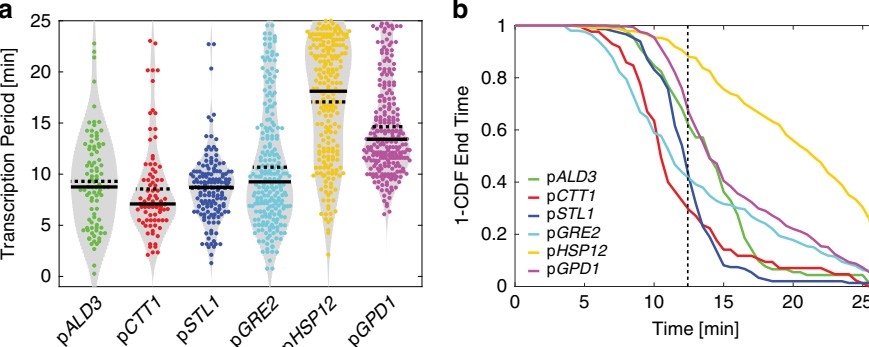

**Fig. 7 Transcription shutoff from different promoters. a** Violin plots representing the Transcription Period measured for the six different promoters following a 0.2 NaCl stress. Each dot represents the value calculated from a single cell. The solid line is the median and the dashed line the mean of the population. **b** One minus the cumulative distribution function of End Times for the different promoters. The vertical dashed line represents the median adaptation time of Hog1 at 0.2 M NaCl.

role played by MAPKs, and Hog1 in particular, on multiple steps of the transcriptional process[47,48].

The ramp experiment starts with a pulse at 0.2 M NaCl followed by a slow increase of the NaCl concentration up to 0.6 M over the next 20 min (Supplementary Movie 6). This constant rise in external osmolarity extends the Hog1 activity window by preventing the adaptation of the cells. More cells can become transcriptionally active and the transcription shut off is delayed (Fig. 6d, e). However, in these conditions, there is a clear lack of correlation between Hog1 activity, which is sustained in many cells over the 30 min of the time-lapse, and the transcription output of the p*STL1* that stops much earlier (Fig. 6f). Taken together, these experiments demonstrate that the MAPK activity is required but not sufficient to sustain the transcriptional process. In the ramp experiment, transcription cannot be sustained throughout the whole Hog1 activity window, demonstrating that other factors contribute to limiting the duration of the transcription.

**Promoter identity influences the transcription shutoff time.** In order to test whether the promoter identity plays a role in the process of transcriptional shutoff, we quantified the duration of the transcriptional period for the six promoters and plotted the cumulative distributions of End Times following a 0.2 M NaCl stress (Fig. 7a, b). Interestingly, despite similar cell volume adaptation times for all the experiments, the promoters display substantially different kinetics of inactivation. Promoters transcribing at a lower level (p*CTT1* and p*GRE2*) terminate transcription earlier. This shorter transcriptional window may reflect an inferior recruitment of transcriptional activators to the promoter, enabling an earlier inhibition of transcription due to chromatin closure. In addition, promoters with basal activity display an extended period of transcription after adaptation (Fig. 7a, b). For p*GPD1* and p*GRE2*, this results in a biphasic decay, where the first phase corresponds to the arrest of Hog1-induced transcription and the second phase can be associated to the basal transcription arising from these promoters (Fig. 7b). Note that basal transcription may even be increased due to a higher basal Hog1 signaling activity post high osmolarity conditions[49].

Remarkably, p*HSP12* transcription persists beyond the adaptation time, with nearly 30% of the cells displaying an active TS at the end of the experiment. This suggests that basal expression from this promoter is strongly increased post-stimulus. In contrast to p*GPD1* and p*GRE2*, p*HSP12* possesses numerous Msn2/4 binding sites. Although the relocation dynamics of Hog1 and Msn2 are highly similar during the adaptation phase, Msn2

displays some stochastic secondary pulses[27] that are not correlated to Hog1 relocation events. This could explain the stronger basal expression arising from this promoter post-adaptation (Supplementary Fig. 1e, f).

To summarize, these measurements demonstrate that the pattern of MAPK activity provides a temporal window where transcription can take place. When the signaling cascade is shut off, transcription ceases soon afterward. However, the promoter identity, and probably its propensity to recruit positive activators, will determine for how long it can sustain an open chromatin environment favorable to transcription before Hog1 activity decreases due to cellular adaptation.

**Discussion**

In this study, we have constructed PP7 reporter strains to monitor the transcription dynamics of osmostress promoters. The second exogenous copy of the promoter is integrated at the *GLT1* locus. This strategy provides a similar genomic environment for all the promoters, in order to compare their specific characteristics. Interestingly, we saw only minor differences in the CDF of Start Times of the p*STL1* when integrated at its endogenous locus compared with the *GLT1* locus. This observation provides a strong evidence that TF binding and chromatin state of the duplicated promoter sequences mimic closely the ones at the native environment of the gene. Note that the signal at the TS is expected to be proportional to the length of the transcribed mRNA. The *GLT1* locus with its 6.5 kB length was expected to provide a signal four times stronger than the endogenous *STL1* ORF (1.7 kB). The unexpectedly high signal obtained from the PP7 reporter at the endogenous locus may be indicative of global difference in transcription rates between the *GLT1* and *STL1* ORFs alternatively, the smaller *STL1* ORF might enhance transcription efficiency via gene-looping[50,51].

Our data illustrate the complex balance that exists between positive and negative regulators taking place at the stress-induced loci. At each locus, positive and negative regulators will control the level and duration of transcription. We have shown that the first few minutes of Hog1 activity are essential to initiate the transcription. Transcription factors, chromatin remodelers such as the SAGA and RSC complexes, together with Hog1 will contribute to open and maintain an accessible chromatin environment at the stress-response loci[35,40]. Once initiated, transcription seems highly processive and only in a small fraction of traces, we are able to detect a pause in transcription. However, it has been shown that PolII recruits additional chromatin remodelers, including the Ino80 complex and Asf1 that will redeposit nucleosomes after acute transcription[52]. These conflicting

activities will determine the overall duration of transcription at a locus. Indeed, promoters with lower transcriptional activity, such as pGRE2 and pCTT1, recruit fewer positive activators and will be repressed faster by the negative regulators.

The repression level of a promoter during log-phase growth will determine the speed and the noise of the transcription activation process. Thus, for each promoter, a trade-off has to be found between these two contradictory requirements. For instance, GPD1, which is essential for survival to osmotic stress, has an important basal expression level and can thus be rapidly induced upon stress. Interestingly, the chromatin state, encoded in part by the promoter sequence, can be tuned by external growth conditions. Thus, the noise generated by a promoter is not rigidly set by its DNA sequence but fluctuates based on the environment.

In higher eukaryotes, the stress response MAPKs p38 and JNK relocate to the nucleus upon activation[53,54]. Early genes such as c-Fos or c-Jun, are induced within minutes after activation of signaling[3,55]. Interestingly, these loci display basal expression and require minimal chromatin modification for their induction[56,57]. Conversely, delayed primary response and secondary response genes require more chromatin remodeling to induce their activation[55,58]. These similarities with the regulation of Hog1-dependent genes induction suggest a high conservation in the mechanisms used by MAPK in eukaryotes to regulate the dynamics of gene expression.

## Methods

**Plasmids and yeast strains.** Plasmids, yeast strains and primers used in this study are listed in Supplementary Tables 1–3. All strains were constructed in the W303 background. Transformations were performed with a standard lithium-acetate protocol. Gene deletions and gene taggings were performed with either pFA6a cassettes[59,60] or pGT cassettes[61]. Transformants were selected with auxotrophy markers (Uracil, Histidine, Leucine, Tryptophan, Adenine) and gene deletions were performed with antibiotic resistance to Nourseothricin (NAT) or Kanamycin (KAN). In order to generate the membrane-anchored Hog1, the pGTT-mCherry vector was modified by inserting annealed oligos (oSP1648/9) encoding the last 30 bp of the Ras2 sequence to obtain the pGTT-mCherry-CaaX plasmid. A strain possessing the Hog1-mCherry:LEU2 modification was transformed with the pGTT-mCherry-CaaX plasmid linearized with XbaI and SacI to induce a marker switch and introduce the membrane anchoring motif.

**PP7 and MS2 strains construction.** The PP7-GFP plasmids are based on the bright and photostable GFPenvy fluorescent protein[32]. The PP7 protein was derived from Larson et al.[30] (Addgene# 35194) with an additional truncation in the capsid assembly domain (PP7ΔFG residues 67–75: CSTSVCGE[31]). The expression of the PP7 construct is controlled by an ADH1 promoter and a CYC1 terminator. The final construct pVW284 was cloned in a Single Integration Vector URA3 (pSIVu[61]). The PP7-mCherry was cloned by replacing the GFP by the mCherry sequence. The MS2-GFP was generated by using the original MS2 sequence from Betrand et al.[18], which also lacks the capsid assembly domain (Addgene# 27117), inserted into the pVW284. The PP7 stem-loops plasmids are based on the previously published pPOL1 24xPP7sl integrative plasmid[30] (Addgene #35196). The stress-responsive promoters replace the POL1 promoter in the original construct using 1 kb (0.8 kb for pSTL1, 0.66 kb for pALD3) upstream of the start codon. The pSTL1-24xMS2sl was generated by replacing the PP7 stem-loops with the MS2 stem-loops obtained from Betrand et al.[18] (Addgene# 31865).

A strain bearing the Hta2-mCherry nuclear marker and expressing the PP7-GFP was transformed with plasmids containing the different osmostress promoters driving the PP7sl production, linearized with a NotI digestion and integrated upstream of the GLT1 ORF, as previously published[30]. Correct integration into the GLT1 locus was screened by colony PCR with primers in the GLT1 ORF (oSP061, +600 bp) and in the TEF terminator (oSP062) of the selection marker on genomic DNA extractions. The integrity of the PP7 stem-loops array was assessed with primers within the TEF terminator (oSP062) and in GLT1 ORF (oVW447, +250 bp) for all the promoters and deletions, after each transformation performed. For all the strains used in the study, at least two clones with correct genotypes were isolated and tested during a salt challenge time-lapse experiment. From the data analysis, the most frequent phenotype was isolated and the strain selected.

To tag the endogenous locus of STL1 with the 24xPP7sl, the plasmid pSP264 with the STL1 promoter was modified by replacing the GLT1 ORF sequence by a 500 bp sequence starting 100 bp after the start codon of STL1. The plasmid was digested SacI-NotI and purified over a gel to isolate a fragment that contains the pSTL1-24xPP7sl-STL1$_{100-600}$. A double-strand break was generated in the STL1 ORF using Cas9 and a sgRNA targeting the PAM motif GGG 62 bp upstream of the start codon. The Cas9

and sgRNA are expressed from a 2 μ plasmid (Addgene #35464[62]) slightly modified from the work from Laughery et al. (Addgene# 67639[63]). The purified DNA fragment containing the stem-loops was used as repair DNA to promote homologous recombination at the STL1 locus (Supplementary Fig. 4a). The correct size of the inserted fragment was verified by colony PCR around the PP7sl insert. Multiple positive clones were screened by microscopy. The results from two transformants are presented in this work to ensure that potentially undesired DNA alterations by Cas9 do not affect the response in the HOG pathway.

In order to generate the diploid reporter strain, a MATa strain containing the PP7-mCherry::URA3, pSTL1 24xPP7sl:GLT1 and Hta2-tdiRFP:TRP1 was crossed to a MATα strain bearing the MS2-GFP::URA3, pSTL1 24xMS2sl:GLT1 and Hta2-tdiRFP:NAT. Haploid cells were mixed on a YPD plate for a few hours before cells were resuspended in water and spread with beads on a selection plate (SD-TRP + NAT).

The plasmids generated for this study are available on Addgene.

**Yeast culture.** Yeast cells were grown in YPD medium (YEP Broth: CCM0405, ForMedium) for transformation or in Synthetic Defined (SD) medium (YNB: CYN3801/CSM: DCS0521, ForMedium) for microscopy experiments. Before time-lapse experiments, cells were grown at least 24 h in log-phase. A saturated over-night culture in SD medium was diluted into fresh SD-full medium to OD$_{600}$ 0.025 in the morning and grown for roughly 8 h to reach OD$_{600}$ 0.3–0.5. In the evening, cultures were diluted by adding (0.5/OD$_{600}$)μl of cultures in 5 ml SD-full for an overnight growth that kept cells in log-phase conditions. Cultures reached an OD$_{600}$ of 0.1–0.3 in the morning of the second day and were further diluted when necessary to remain below an OD$_{600}$ of 0.4 during the day. To prepare the samples, these log-phase cultures were further diluted to an OD$_{600}$ 0.05 and sonicated twice 1 min (diploids were not sonicated) before placing 200 μl of culture into the well of a 96-well glass-bottom plate (MGB096-1-2LG, Matrical Bioscience) previously coated with a saturated solution of Concanavalin A diluted to 0.5 mg/ml in water (C2010, Sigma-Aldrich)[64]. Cells were let to settle for 30–45 min before imaging. Osmotic shock was performed under the microscope, by adding 100 μl of a three times concentrated SD-full+NaCl stock solutions to the 200 μl of medium already in the well, to reach the final desired salt concentration.

**Microscopy.** Images were acquired on a fully automated inverted epi-fluorescence microscope (Ti2-Eclipse, Nikon) placed in an incubation chamber set at 30 °C. Excitation was provided by a solid-state light source (SpectraX, Lumencor) and dedicated filter sets were used to excite and detect the proper fluorescence wavelengths with a sCMOS camera (Flash 4.0, Hamamatsu). A motorized XY-stage was used to acquire multiple fields of views in parallel and a piezo Z-stage (Nano-Z200, Mad City Labs) allowed fast Z-dimension scanning. Micro-manager was used to control the multidimensional acquisitions[65].

Experiments with PP7 stem-loops were acquired with a 60X oil objective. For strains with PP7-GFP and Hta2-mCherry, GFP (40 ms, 3% LED power) and RFP (20 ms), along with two bright field images were recorded every 15 s for the GFP and every minute for the other channels, for a total duration of 25 min. Six z-stacks were performed on the GFP channels covering ±1.2 μm from the central plane with 0.4 μm steps. An average bleaching of 32% for the GFP and 26% for the RFP for the whole time-lapse was quantified in a strain without the PP7 stem-loops, to avoid artifacts from the appearance of bright fluorescent foci. For all time-lapse experiments, media addition was performed before time point 4, defined as time zero. All microscopy experiments were performed in duplicate for non-induced control experiments and at least triplicate for the NaCl induced experiments.

**Flow chamber experiments.** The flow experiments were performed in Ibidi chambers (μ-Slide VI 0.4, Ibidi). Two 50 ml Falcon tube reservoirs containing SD-full +0.5 μg/ml fluorescein-dextran (D3305, ThermoFischer) and SD-full +0.6 M NaCl were put under a pressure of 30 mbar (FlowEZ, Fluigent). The media coming from each reservoir were connected using FEP tubing (1/16″ OD × 0.020″ ID, Fluigent) to a 3-way valve (2-switch, Fluigent). The concentration of NaCl in the medium was controlled using a Pulse-Width Modulation strategy[66,67]. Periods of 4 s were used and within this time, the valve controlled the fraction of time when SD-full versus SD-full + NaCl was flowing. TTL signals generated by an Arduino Uno board and dedicated scripts were used to control precisely the switching of the valve. The fluorescein present in the SD-full medium quantified outside the Cell object provided an estimate of the NaCl concentration in the medium. Some strong fluctuations in this signal were probably generated by dust particles in the imaging oil or FLSN-dextran aggregates in the flow chamber. Following 24 h log-phase growth, cells bearing the pSTL1-PP7sl reporter, Hog1-mCherry, and Hta2-tdiRFP tags were diluted to OD 0.2, briefly sonicated and loaded in the ibidi channel previously coated by Concanavalin A. Cells were left to settle in the channel for 10 min before SD-full flow was started.

**Raffinose experiment.** For the experiments comparing pSTL1-PP7sl induction in glucose versus raffinose, cells were grown overnight to saturation in SD-full medium. The cultures were diluted to OD 0.025 (Glucose) or 0.05 (Raffinose) and grown at 30 °C for at least four hours. In the raffinose medium, the expression level of the PP7-GFP was twofold lower than in glucose. Because of this low fluorescence

intensity, cells were imaged with a 40X objective, and a single Z-plane was acquired. Manual curation of the images was performed to define the Start Time in more than 250 cells. This experiment was performed in duplicate.

**Data analysis**. Time-lapse movies were analyzed in an automated way: cell segmentation, tracking, and feature measurements were performed by the YeastQuant platform[33] based on Matlab. Summary of the dataset, strains, and cell numbers are provided in Supplementary Table 4. All PP7 experiments were realized in at least two or three fully independent replicate experiments. A representative experiment was selected for each strain and inducing conditions, based on cell size and cell adaptation dynamics. The replicates which did not pass one of these controls were discarded from the replicate analyses. Individual single-cell traces were filtered based on cell shape and GFP intensity to remove segmentation errors or other experimental artifacts. In addition, cells in mitosis were removed from the analysis with a 0.95 filter on the nuclei eccentricity, to remove artifacts from locus and PP7 signal duplication.

The Hta2 signal combined with the two bright field images allowed to define the nucleus and cell borders. The GFP z-stacks were converted by a maximum intensity projection in a single image that was used for quantification. In order to avoid improper quantification of transcription sites at the nuclear periphery, the Nucleus object defined by the histone fluorescence was expanded by 5 pixels within the Cell object to define the ExpNucl object.

Further analysis was performed by dedicated Matlab scripts. The transcription site intensity was quantified by the difference between the mean intensity of the 20 brightest pixels (HiPix) in the ExpNucl and the median intensity from the whole cell. This provides a continuous trace that is close to zero in absence of TS and increases by up to few hundred counts when a TS is present. To identify the presence of a transcription site, a second feature named ConnectedHiPix was used (Supplementary Fig. 2a). Starting from the 20 HiPix, a morphological opening of the image was performed to remove isolated pixels and retaining only the ones that clustered together which correspond to the transcription site. The ConnectedHiPix value was set to the mean intensity of the pixel present in the largest object remaining after the morphological operation. If no pixel remained after the morphological operation, the ConnectedHiPix was set to "NaN". In each single-cell trace, ConnectedHiPix values only detected for a single time point were removed. After this filtering, the first and last time points where a ConnectedHiPix was measured were defined as transcription initiation (Start Time) and shutoff (End Time), respectively. Manual curation of Start and End Times from raw microscopy images was performed to validate this transcription site detection strategy (Supplementary Fig. 2b, c). In order to detect individual transcriptional bursts in the HiPix traces, the *findpeak* algorithm was used to identify in the trace all the peaks larger than a threshold of seven counts within the Start and End Times. Following this first process, a set of conditions were defined to retain only the more reliable fluctuations: the drop following the peak has to be larger the fourth of the peak intensity; the intensity of the following peak has to rise by more than a third of the value at the trough. In addition, the value of the peak has to be at least one-fifth of the maximum intensity of the trace in order to remove small intensity fluctuations being considered as peaks.

**Reporting summary**. Further information on research design is available in the Nature Research Reporting Summary linked to this article.

## Data availability
The raw images and additional features measurements that support the findings of this study are available from the corresponding author upon reasonable request. All other relevant data supporting the key findings of this study are available within the article and its Supplementary Information files or from the corresponding author upon reasonable request. The source data for Figs. 1d, 2a, f, g, 3a, 4b, c, and 6c and Supplementary Figs. 1, 4–6, 8, 12, and 13 are provided as Source Data file. A reporting summary for this article is available as a Supplementary Information file.

## Code availability
A general description of the image analysis platform has been published previously[33]. A more recent version of the code has been made available of GitHub [https://github.com/sergepelet/YeastQuantX_V2]. A script to extract measured parameters from the data is provided as a supplementary file.

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

## Acknowledgements

We thank the members of the Pelet lab and Martin lab and for helpful discussions. Marta Schmitt, Yves Dusserre, Gaëlle Spack, Joan Jordan, and Clémence Varidel for technical assistance. David Shore and his lab for helpful discussions and reagents. Marie-Pierre Peli-Gulli and Claudio de Virgilio for plasmids, Tineke Lenstra for suggesting the PP7ΔFG allele. Agathe Pelet for manual curation of microscopy images. Eulalia de Nadal, Mariona Nadal-Ribelles, and Veneta Gerganova for critically reading the manuscript. Work in the Pelet lab is supported by SystemsX.ch (IPhD 51PHP0_157354), the Swiss National Science Foundation (SNSF, PP00P3_172900 and 31003A_182431), and the University of Lausanne.

## Author contributions

VW and SP designed the experiments, analyzed the data, and wrote the manuscript. VW established the conditions for the PP7 imaging. VW and SP performed the experiments.

## Competing interests

The authors declare no competing interests.
