## [Peer Review File · Nature Communications]

Reviewers' comments:

Reviewer #1 (Remarks to the Author):

In this article, the authors report a quantitative analysis of transcription dynamics induced by osmotic stress in the yeast *S. Cerevisiae*. They monitored the accumulation of nascent mRNA at transcription sites following osmotic activation of 6 promoters which are known to be upregulated following an osmotic stress (pALD3, pCTT1, pSTL1, pGRE3, pHSP12, pGPD1). To do so they use a PP7 reporter assay: constitutively expressing PP7-GFP and replacing the promoter of GLT1 by the promoter of interest followed by 24 repeats of PP7sl. Transcription sites are then visible in time lapse microscopy through the GFP channel and an automated image analysis pipeline allows to quantify the transcription dynamics of these promoters. The dataset is of high quality and may bring a quantitative description of the dynamics of these promoters (see below). In particular, the existence of different transcription dynamics for the set of genes that are repressed in basal conditions and the ones that show a low level of expression is very interesting for our understanding of stress responsive genes in general.

I have, however, two major concerns regarding this study.

First, I think that the authors should make clearer the conclusions of their study and discuss more the relevance of their study for biology: specific conclusions are not included in the abstract and in the main conclusions, there is only a short general discussion in the last paragraph (line 452-461). Second, I would like more details as for why the authors choose to study transcription at a different locus than the native locus of each genes of interest. It is known that transcription is altered by gene positioning, in particular for genes such as STL1 which are placed in subtelomeric region. Of course, by placing the promoters of interest at the same locus they can study the impact of the promoter sequence only, and disregard the chromatin context, but this means also that the results are not necessarily relevant to understand the physiological response of yeast cells to osmotic stress. I am left wondering to what extent the observed dynamics would be identical if the PP7sl repeats were placed at the native locus of each genes. This question of gene positioning is of special importance in the case of the osmotic stress response which is, as recalled by the authors, accompanied by chromatin remodeling. Of note, the native genes are not deleted from the strain, and therefore there are two copies of the promoter of interest, one at the native locus driving the gene of interest, and another one at the GLT1 locus driving the PP7sl repeats. This should be discussed in the article, in particular regarding the comparison of dynamics of Hog1 nuclear localization and transcription dynamics (line 126+). Ideally, the authors should check that transcription occurs at both locus and with the same dynamics.

Minor comments:

Line 125 : dPSTR. Explain briefly (cf. Nature Communications volume 7, Article number: 11304 (2016))

Figure 1A: remove the "C" within the panel A (just below Asf1 protein).

Figure 1D: the shaded areas (S.E.M) are difficult to see. It could be interesting also to show the variability of the response by showing the standard deviation as a shaded area.

Figure 2A: the shaded areas (S.E.M) are difficult to see

Figure 6F: the authors mention that they observed synchrony between PP7 end time and Hog1 adaptation time only for the "pulse" experiment. I agree that both events occur within the same

timing, but the fact that all data points for the pulse experiment collapse to the same location prevent any conclusions on the correlation between these two events.

Reviewer #2 (Remarks to the Author):

This manuscript could be seen as a continuation of a problem previously addressed by the corresponding author. It relates to the question how a transient signaling system organizes a transcriptional response and how much variance can be expected if one looks at individual, genetically identical cells. For the yeast osmo-stress response this has been already attempted and accomplished by several groups leading to the insight that there is indeed a surprisingly large variation with regard to the transcriptional dynamics that can also exhibit a high and variable background noise depending on the individual promoter studied. The current work extends these previous studies by using a reporter system that allows high temporal resolution, reliable and sensitive quantification and a continuous observation covering start, peak and shut down of individual transcription events. Moreover, the way the system is constructed one can compare the behavior of different gene promoters without considering effects on elongation and termination by the size and sequence of the coding regions of these genes. The experiments yielded some interesting observations and rules that have clearly important considerations for similar systems studied in higher eukaryotes. Of course they are also most interesting for the yeast community studying signal transduction systems and transcription.

The study is extensive and even if one might easily think of additional questions and experiments I would consider the data sets complete and with enough detail that should warrant publication as is. (If I had any intention to ask for more experiments I might ask why the authors stopped at 0.3 M NaCl stress and did not go to higher stress concentrations to see whether the starting times and the quantitative response of the different promoters under study would then lose their differences). All the data and their statistical analysis are of high quality throughout. The validity of the approach is clearly and extensively documented in the supplementary part. However, what I would like to know is whether and how often, to what detail the approach based on the Larson et al. (Singer lab) publication has been successfully used by other labs and problem areas. Perhaps the authors should mention that. Otherwise the references are pretty much complete.

One suggestion with respect to the manuscript might be that one could try to make it slightly shorter by avoiding some redundancies between result and discussion section. In addition one could perhaps see whether a slightly longer title could raise the interest in this study.

Reviewer #3 (Remarks to the Author):

Wosika and Pelet studied at single cell level the response of yeast cells to osmotic upshock regimes. Specifically, they used gene expression reporters to monitor transcription from different osmostress-induced genes. The system is set up such that mRNAs are detected at the site of production and hence the system provides readouts for transcription and the load of RNA polymerases. While osmostress responses have been studied extensively in yeast, also at single cell level, the monitoring

of transcription as described here is novel and adds a new dimension to the well-studied yeast HOG signalling system.

The (rich) data generated in the study lead to several interesting observations. For instance, it appears that Hog1 can initiate transcription of target genes only during the initial phases when nuclear Hog1 increases and reaches its maximum, not when it declines. The data also confirm earlier observations that genes that have basal expression will be induced faster and in a higher proportion of cells than genes that are completely shut off under non-inducing conditions. Finally, the data also show that different target genes show somewhat different temporal behaviour, which may be attributed to the specific promoter architecture.

While each of these observations is potentially interesting the manuscript remains at a rather descriptive level. Yeast strains lacking certain transcription factors were studied and some potentially underlying mechanisms are discussed but not studied in the present manuscript. Still, the data presented here open up for a possibility to study gene transcription at a quantitative and temporal resolution in a tractable experimental system that offers opportunities for groups with specific interest in transcriptional regulation.

Overall the system is designed in an elegant way and data collection and data management seem appropriate. I also believe that the validation of the system is well done. This said, in order to demonstrate that the observed effects are truly Hog1-dependent (which they almost certainly are), a control with cytoplasmically tethered Hog1 could have been performed. Such constructs are available from the Thorner lab.

Why were deletion of HOT1 and SKO1 chosen and not MSN2 MSN4?

Stefan Hohmann

We thank the reviewers for taking the time to read and comment on our manuscript. We appreciate the constructive suggestions that they have provided in order to improve this work. You will find below the main changes and additions that we included in the revised version of the manuscript. On the next, pages, a point-by-point response to each reviewer's comment is provided.

General revisions

1. We have engineered a strain where the endogenous *STL1* locus was targeted with the PP7 stem loops. This strain allowed us to verify that the promoter at its native locus and at the *GLT1* locus behaved in an extremely similar manner. These results thus validate our strategy of using a 0.7 to 1kB sequence upstream of the gene inserted in front of the PP7sl in the *GLT1* locus to study the differences in stress-induced promoters.
2. We have generated a diploid strain combining a MS2-GFP and a PP7-mCherry reporters together with PP7 and MS2 stem loops controlled by two *STL1* promoters inserted identically at the *GLT1* loci on the two chromosomes IV. This dataset allows to better visualize the noise in gene expression in a single cell.
3. A membrane-anchoring motif was attached to the MAPK Hog1 in order to limit its capacity to induce transcription. The induction of the *STL1* and *GPD1* promoters were studied under these conditions with very different outcomes. While p*STL1* induction was almost completely abolished, p*GPD1*-induced transcription was barely affected.
4. The stress-responsive transcription factors Msn2 and Msn4 were both deleted in strains bearing the p*STL1*-PP7 or the p*GPD1*-PP7 reporters. Only small changes in the transcriptional output from these two promoters were observed in these mutants which confirms that p*STL1* and p*GPD1* are specific Hog1 targets.
5. Source data is provided for all main figures and relevant supplementary figures. Due to the large size of the raw imaging datasets, only the PP7 foci measurements are provided. The original raw data and processed data will be made available upon request.
6. The deposition of all PP7 and MS2 plasmids at Addgene is under way such that the constructs can be available upon publication of the manuscript.
7. Finally, we have removed redundancies between the Discussion and the Results sections, improved the abstract and discussion and modified the title.

Point-by-Point Response

Reviewer #1 (Remarks to the Author):

*In this article, the authors report a quantitative analysis of transcription dynamics induced by osmotic stress in the yeast *S.Cerevisiae*. They monitored the accumulation of nascent mRNA at transcription sites following osmotic activation of 6 promoters which are known to be upregulated following an osmotic stress (pALD3, pCTT1, pSTL1, pGRE3, pHSP12, pGPD1).*

To do so they use a PP7 reporter assay: constitutively expressing PP7-GFP and replacing the promoter of GLT1 by the promoter of interest followed by 24 repeats of PP7sl. Transcription sites are then visible in time-lapse microscopy through the GFP channel and an automated image analysis pipeline allows to quantify the transcription dynamics of these promoters. The dataset is of high quality and may bring a quantitative description of the dynamics of these promoters (see below). In particular, the existence of different transcription dynamics for the set of genes that are repressed in basal conditions and the ones that show a low level of expression is very interesting for our understanding of stress responsive genes in general.

I have, however, two major concerns regarding this study.

First, I think that the authors should make clearer the conclusions of their study and discuss more the relevance of their study for biology: specific conclusions are not included in the abstract and in the main conclusions, there is only a short general discussion in the last paragraph (line 452-461).

We have changed the abstract and the discussion to clarify the main findings of this paper.

Second, I would like more details as for why the authors choose to study transcription at a different locus than the native locus of each genes of interest. It is known that transcription is altered by gene positioning, in particular for genes such as STL1 which are placed in subtelomeric region. Of course, by placing the promoters of interest at the same locus they can study the impact of the promoter sequence only, and disregard the chromatin context, but this means also that the results are not necessarily relevant to understand the physiological response of yeast cells to osmotic stress. I am left wondering to what extent the observed dynamics would be identical if the PP7sl repeats were placed at the native locus of each genes. This question of gene positioning is of special importance in the case of the osmotic stress response which is, as recalled by the authors, accompanied by chromatin remodeling. Of note, the native genes are not deleted from the strain, and therefore there are two copies of the promoter of interest, one at the native locus driving the gene of interest, and another one at the GLT1 locus driving the PP7sl repeats. This should be discussed in the article, in particular regarding the comparison of dynamics of Hog1 nuclear localization and transcription dynamics (line 126+). Ideally, the authors should check that transcription occurs at both locus and with the same dynamics.

In this study, we have constructed our strains by adding a second copy of the promoter of interest upstream of the *GLT1* ORF which is 6,5 kB long and encodes a non-essential protein. Indeed, as nicely described by Ferraro et al. (Ferraro, *WIREs Dev Biol* 2016), the integration site (5'UTR versus 3'UTR) influences considerably the readout of phage coat protein-based assays, due to the direct correlation between the time that the RNA Pol II spends on the locus after the transcription of the stem loops and the transcription site intensity measured. The six different promoters chosen for this study encode transcripts of highly heterogenous lengths, from 330 bp (*HSP12*) to 1710 bp (*STL1*). This up to five-fold difference in transcript lengths would greatly influence the fluorescence intensities measured and thus prevent a direct comparison of the promoter induction.

In addition, the transcript containing the twenty-four stem-loops cannot be translated, due to the presence of multiple start and stop codons within the loops. Insertion of the stem loops upstream of a gene corresponds to a null mutant of that gene. While *glt1Δ* have very few deficiencies and no known phenotype upon hyper-osmotic stress (SGD), this is not the case for our genes of interest. For instance, placing the loops upstream of the *GPD1* locus would strongly perturb the response of the cells to hyper-osmotic challenge, since it is essential for the growth under these conditions (Albertyn MBoC 1994).

Regarding the presence of an extra-copy of a promoter in the genome, we believe that it creates only a minimal perturbation to the general stress response of the cells and the transcriptional program induced upon osmotic challenge. Roughly 300 genes are induced upon activation of the HOG pathway and integration of one additional target is unlikely to result in a large change in this response.

Taken together, we believe that these arguments validate our choice of using the *GLT1* locus as a common landing platform for all our reporters, a strategy that was previously used in the initial PP7 paper by Larson and colleagues (Larson, Science 2012). However, we certainly agree with the reviewer that the location of a gene is important and that the chromatin state surrounding the gene can modulate the expression output of a promoter. When building these strains, we do not make the assumptions that the chromatin is identical to the native locus, however, it has been shown by numerous studies in yeast that the 1kB upstream of the Start codon is generally sufficient to recapitulate to a great extent the promoter expression dynamics.

In order to verify this assumption, we have built an endogenously tagged p*STL1* 24xPP7*sl* reporter strain and compared it to the *GLT1* integrated reporter (new Supplementary Fig. 4). Because the *STL1* transcript is expected to be about four times smaller than the *GLT1* one, we were expecting to measure significantly lower TS intensities which would limit our ability to detect responding cells. To our surprise, the mRNA transcription characteristics measured at the endogenous locus are very close to the ones obtained at the *GLT1* locus. Importantly, the distribution of Start Time measured at the two loci are almost identical suggesting that the regulation of transcription activation takes place with similar kinetics on the two promoters. The percent of responding cells, and the trace intensity are slightly lower at the endogenous locus while transcription shutoff takes place 2 min earlier at the native locus. The high intensity measured at the *STL1* locus suggests that either transcription speed of the *STL1* ORF is very different from the one on the *GLT1* ORF, alternatively, the shorter ORF and the *STL1* terminator might favor gene-looping, leading to an increased loading of the PolIII on the *STL1* ORF. Further studies will be needed to assess the importance of this phenomenon.

Unfortunately, we cannot directly compare the transcription from the *STL1* and the *GLT1* ORF in the same cell as suggested by the reviewer. Experiments performed with a diploid strain combining a PP7 and MS2 reporter demonstrate the lack of correlation in the activity of two p*STL1* at the *GLT1* locus in the same cell (New Supplementary Fig. 6).

Minor comments:

Line 125 : dPSTR. Explain briefly (cf. Nature Communications volume 7, Article number: 11304 (2016))

We have added the description of the dPSTR system in the main text and in the Supplementary Figure legend.

Figure 1A: remove the "C" within the panel A (just below Asf1 protein).

Thank you for noticing this.

Figure 1D: the shaded areas (S.E.M) are difficult to see. It could be interesting also to show the variability of the response by showing the standard deviation as a shaded area.

Figure 2A: the shaded areas (S.E.M) are difficult to see

We have tried to improve this by decreasing the transparency of the SEM area. We agree that showing the standard deviation or the 25- 75- percentiles would provide an interesting additional information. Unfortunately, when combining a large number of curves such as in Figure 2A it becomes extremely difficult to sort out the various traces on the graph. Therefore, we decided to use the SEM areas throughout the manuscript for the PP7 intensity graphs. In order to better demonstrate the variability of the induction of the pSTL1 promoter, we have added a Supplementary Figure 6 that displays the variability in activation of two pSTL1 promoters in diploid driving either MS2 or PP7 stem loops. The lack of correlation between the Start Time and the maximum of the traces are good illustrations of the intrinsic noise generated by this promoter.

Figure 6F: the authors mention that they observed synchrony between PP7 end time and Hog1 adaptation time only for the "pulse" experiment. I agree that both events occur within the same timing, but the fact that all data points for the pulse experiment collapse to the same location prevent any conclusions on the correlation between these two events.

We present below a blown-up version of the figure 6F. And even in this figure, the fact that we have specific imaging time points limits the resolution of the graph and some single cell measurements fall on the same spot. In the graph below data points combining to multiple single cell measurements are displayed with a more intense green color. However, the main message from this pulse experiment is that quickly after Hog1 activity stops, mRNA transcription is also suspended. If MAPK activity was necessary only for the initiation of transcription, we would have observed PP7 End Time with value up to 20 min, similarly to what we observe in the step experiment.

Reviewer #2 (Remarks to the Author):

This manuscript could be seen as a continuation of a problem previously addressed by the corresponding author. It relates to the question how a transient signaling system organizes a transcriptional response and how much variance can be expected if one looks at individual, genetically identical cells. For the yeast osmo-stress response this has been already attempted and accomplished by several groups leading to the insight that there is indeed a surprisingly large variation with regard to the transcriptional dynamics that can also exhibits a high and variable back ground noise depending on the individual promoter studied. The current work extends these previous studies by using a reporter system that allows high temporal resolution, reliable and sensitive quantification and a continuous observation covering start, peak and shut down of individual transcription events. Moreover, the way the system is constructed one can compare the behavior of different gene promoters without considering effects on elongation and termination by the size and sequence of the coding regions of these genes. The experiments yielded some interesting observations and rules that have clearly important considerations for similar systems

studied in higher eukaryotes. Of course they are also most interesting for the yeast community studying signal transduction systems and transcription.

The study is extensive and even if one might easily think of additional questions and experiments I would consider the data sets complete and with enough detail that should warrant publication as is. (If I had any intention to ask for more experiments I might ask why the authors stopped at 0.3 M NaCl stress and did not go to higher stress concentrations to see whether the starting times and the quantitative response of the different promoters under study would then lose their differences).

We have limited our dose-responses from 0.0 to 0.3M NaCl in order to remain at concentrations where only a minimal delay due to cell shrinkage is observed. For

example, in Figure 1d, a small delay at 0.3M is already noticeable. As NaCl concentration increases, cellular crowding induces a lag in the cellular and transcriptional response from the cells (Miermont PNAS 2013, Aymoz Nat. Comm. 2016). In addition, cell shrinkage generates some experimental artifact in the measured fluorescence intensity (small initial peak at time zero) that could perturb the detection of PP7 foci at higher concentrations of NaCl.

All the data and their statistical analysis are of high quality throughout. The validity of the approach is clearly and extensively documented in the supplementary part. However, what I would like to know is whether and how often, to what detail the approach based on the Larson et al. (Singer lab) publication has been successfully used by other labs and problem areas. Perhaps the authors should mention that. Otherwise the references are pretty much complete.

In the literature both the MS2 and the PP7 system have been used to monitor mRNA in live cells. Although less used than the MS2 system, the PP7 system has been shown to offer a stronger binding to the mRNA stem loop and thereby a better detection efficiency (Wu Biophys. J. 2012). In higher eukaryotic cells, introns are sometimes used as integration targets to reduce the effect of the stem loop insertion in native transcripts (Fukaya Cell 2016). However, in yeast cells, due to their low prominence of introns, endogenous genes are tagged at their 5'UTR if they are not essential (Donovan, EMBO J. 2019) or an additional copy of the promoter is placed at another genomic location such as *GLT1* or *MDM1* (Larson Science, 2011 or Rullan Mol. Cell 2018). Tagging in the 3'UTR reduced the half-life of the TS fluorescence and its brightness which causes less reliable measurement and required higher imaging conditions and thus more bleaching and possibly phototoxicity.

Nowadays, less invasive approaches have been proposed using RNA aptamer, notably the Spinach and Mango assays (Paige Science 2011, Autour Nat. Comm. 2018). However, to date, none of these assays can reach the brightness of the phage coat protein-based reporter systems. Reviews on the application panels of phage coat proteins can be found here: Buxbaum Nat. Rev. Mol. Cell Biol. 2015 and Urbanek RNA Biol. 2014. We have added some additional references to the manuscript.

One suggestion with respect to the manuscript might be that one could try to make it slightly shorter by avoiding some redundancies between result and discussion section. In addition one could perhaps see whether a slightly longer title could raise the interest in this study.

We have shortened the manuscript length and removed some redundancies between the results and discussion sections. We have changed the title for a longer and hopefully more appealing one.

Reviewer #3 (Remarks to the Author):

Wosika and Pelet studied at single cell level the response of yeast cells to osmotic upshock regimes. Specifically, they used gene expression reporters to monitor transcription from different osmostress-induced genes. The system is set up such that mRNAs are detected at the site of production and hence the system provides readouts for transcription and the load of RNA polymerases. While osmostress responses have been studied extensively in yeast, also at single cell level, the monitoring of transcription as described here is novel and adds a new dimension to the well-studied yeast HOG signalling system.

The (rich) data generated in the study lead to several interesting observations. For instance, it appears that Hog1 can initiate transcription of target genes only during the initial phases when

nuclear Hog1 increases and reaches its maximum, not when it declines. The data also confirm earlier observations that genes that have basal expression will be induced faster and in a higher proportion of cells than genes that are completely shut off under non-inducing conditions. Finally, the data also show that different target genes show somewhat different temporal behavior, which may be attributed to the specific promoter architecture.

While each of these observations is potentially interesting the manuscript remains at a rather descriptive level. Yeast strains lacking certain transcription factors were studied and some potentially underlying mechanisms are discussed but not studied in the present manuscript. Still, the data presented here open up for a possibility to study gene transcription at a quantitative and

temporal resolution in a tractable experimental system that offers opportunities for groups with specific interest in transcriptional regulation.

Overall the system is designed in an elegant way and data collection and data management seem appropriate. I also believe that the validation of the system is well done. This said, in order to demonstrate that the observed effects are truly Hog1-dependent (which they almost certainly are), a control with cytoplasmically tethered Hog1 could have been performed. Such constructs are available from the Thorner lab.

We have added a CaaX motif to the Hog1-mCherry in the strains containing the p*STL1* and the p*GPD1* PP7 reporters. This experiment uncovers another interesting difference between these two promoters. While p*STL1* requires the nuclear enrichment of Hog1 in order to be activated, p*GPD1*-induced transcription is only barely affected by the anchoring of the MAPK to the membrane. These data were added to the new Supplementary Fig. 13 and are discussed in the text.

It has been shown previously that *GPD1* transcription is strongly reduced in *hog1Δ* cells (Rep . Mol. Cell. Biol. 1999). In the original Hog1-CaaX paper from Thorner (Westfall, PNAS, 2008), the authors report a complete abolition of *STL1* mRNA production in the membrane anchored mutant, while *GPD1* mRNA output is reduced by roughly 70%. It remains unclear why we don't observe this amount of reduction in *GPD1* transcriptional output with our assay. However, it could explain why the Hog1-CaaX cells are surprisingly resistant to hyper-osmotic shock.

Why were deletion of HOT1 and SKO1 chosen and not MSN2 MSN4?

Based on the study of Capaldi (Capaldi Nat Gen. 2008), it seemed that deletions of HOT1 and SKO1 would be the best candidates to affect the transcriptional response from p*GPD1* and p*STL1*. Nonetheless, following the reviewer's suggestion, we have generated a strain where both *MSN2* and *MSN4* were deleted. As shown in the new Supplementary Fig. 12, in a *msn2Δmsn4Δ* strain, only minor differences in p*GPD1* Start Time and number of responding cells are observed, in accordance with the population measurements by Rep and colleagues (Rep . Mol. Cell. Biol. 1999). Interestingly, p*STL1* transcription displays an improved transcriptional output in this double mutant. One putative explanation for this behavior is that since many stress induced genes controlled by Msn2 and Msn4 cannot be activated in this background, more PolII and chromatin remodeling factors can be recruited at the *STL1* promoter to induce its transcription.

REVIEWERS' COMMENTS:

Reviewer #1 (Remarks to the Author):

The changes made by the authors and their rebutal letter brought significant and appropriate answers to the question I raised in my initial report. I recommend this article for publication in Nature Communications.

Reviewer #2 (Remarks to the Author):

Very nice paper

Reviewer #3 (Remarks to the Author):

The referee comments have been dealt with in a fully satisfactory manner.